# Flies as Vectors and Potential Sentinels for Bacterial Pathogens and Antimicrobial Resistance: A Review

**DOI:** 10.3390/vetsci9060300

**Published:** 2022-06-16

**Authors:** Ji-Hang Yin, Patrick John Kelly, Chengming Wang

**Affiliations:** 1Department of Pathobiology, College of Veterinary Medicine, Auburn University, Auburn, AL 36849, USA; jzy0089@auburn.edu; 2Department of Veterinary Clinical Sciences, Ross University School of Veterinary Medicine, Basseterre 00265, Saint Kitts and Nevis; pkelly@rossvet.edu.cn

**Keywords:** flies, bacterial pathogens, antimicrobial-resistance, sentinel, surveillance

## Abstract

The unique biology of flies and their omnipresence in the environment of people and animals makes them ideal candidates to be important vectors of antimicrobial resistance genes. Consequently, there has been increasing research on the bacteria and antimicrobial resistance genes that are carried by flies and their role in the spread of resistance. In this review, we describe the current knowledge on the transmission of bacterial pathogens and antimicrobial resistance genes by flies, and the roles flies might play in the maintenance, transmission, and surveillance of antimicrobial resistance.

## 1. Introduction

Flies are insects in the order Diptera that have one pair of wings for flight and a residual second pair of wings, known as knobs, which are used for balance [1]. Over 125,000 species of Dipterans have been identified, including gnats, midges, mosquitoes, leaf miners, horse flies, houseflies, blowflies, and fruit flies. Houseflies, *Musca domestica* Linnaeus (Diptera: Muscidae), are of particular importance as they are notorious “pests” that can transmit a variety of bacterial pathogens [2]. They are thought to have originated in the savannahs of Central Asia and later spread worldwide [3], particularly in tropical and subtropical areas where they are mostly associated with people and domestic animals in both rural and urban areas [4].

Flies have four life stages—eggs, larvae/maggots, pupae, and adults [5]. Female houseflies lay their eggs in compost, trash, soiled bedding, or manure containing moist and microbial-rich decaying organic matter near people’s houses and farms [2]. Each female can oviposit four to six times in her lifetime, each time producing 100–150 eggs [2,6]. The eggs usually hatch within 8–12 h if the environment is moist and at an optimal temperature of 25 °C to 30 °C [2]. The first-instar larvae feed on bacteria in nutrient-rich environments and pass through a further two instars before becoming larvae/maggots that migrate into a dark, dry, and cool place where they pupate [2,6]. Adult flies emerge around 2–4 days later when ambient temperatures are 32 °C to 37 °C, meaning the entire life cycle of houseflies is very rapid, ranging from 10–21 days [7].

The behavior of houseflies promotes their ability to transmit bacterial pathogens [8,9]. They live in close proximity to people (synanthropic) or in their dwellings (endophilic cosmopolitan), and they often feed on animal and human feces (coprophagic) and decaying matter, such as garbage [7]. All life stages can thus be exposed to a variety of pathogens in unsanitary environments, and these can then be mechanically transmitted to people [10]. Adult flies can move over distances of up to 20 miles in their lifetime, which means they are ubiquitous in the environment and well capable of disseminating pathogens from unsanitary areas into people’s homes and places of work and leisure [11].

## 2. Flies as Vectors of Bacterial Pathogens

Flies can carry a surprising diversity and number of pathogens. One systemic review revealed more than 130 human pathogens have been identified in houseflies [3], including bacteria, fungi, viruses, and parasites [3,12] The predominant pathogens are bacteria, including *Klebsiella* spp. [13], *Salmonella* [14], *Pseudomonas aeruginosa* [15], *Campylobacter jejuni* [16], *Edwardisella* spp. [17], *Clostridium* spp. [18], *Yersinia enterocolitica* [19], and *Burkholderia pseudomalliei* [20]. Recently, Balaraman et al. reported that houseflies acquired and harbored infectious SARS-CoV-2 for up to 24 h post-exposure. They could mechanically transmit SARS-CoV-2 genomic RNA to the surrounding environment for up to 24 h post-exposure [12].

### 2.1. External Carriage of Bacterial Pathogens

Flies have unique body structures that enable them to effectively carry bacteria. For example, bacteria readily become attached to the sticky leg pads, hairs, electrostatically charged exoskeleton, and sponging mouthparts of flies [8,21,22]. A study quantifying the transfer of fluorescence-labeled *E. coli* from sugar, milk, steak, and potato salad to houseflies revealed a single housefly can carry up to 2 × 10^12^
*E. coli* and approximately 0.1 mg of food between landing sites [23]. Flies were also found to externally carry *Enterococcus faecium* in poultry farms [24], *Klebsiella pneumoniae* in kitchens and farms [25], *Salmonella enterica* in swine farms [26], and *Staphylococcus aureus* in urban areas [19]. Female flies carry more bacteria than males because they visit oviposition sites that are heavily contaminated with bacteria [27].

### 2.2. Internal Carriage of Bacterial Pathogens

Although most studies have focused on bacteria carried on the body surface, some researchers have investigated bacteria carried internally within the digestive tract [28,29,30,31] Ingested material containing bacteria is initially stored in the crop, from where it passes down the digestive tract through the proventriculus/foregut, midgut, hindgut, and rectum [32]. Whereas epithelial cells in the foregut and hindgut are covered with a protective cuticle [33], the midgut is lined with a unique structure, named the “peritorphic matrix”, peritrophic envelope, or peritrophic membrane (PM) [29,34] (Figure 1). This is a double-layered, noncellular structure composed of chitin, proteoglycans, and various proteins [33] that serves as a physical barrier to prevent microbes in the ingesta from invading epithelial cells and causing damage [34,35]. The PM has gaps, ranging from 2 to 10 nm, which enable digestive enzymes, acid, and secretions to enter the endoperitrophic space and digest food materials [34,36]. At the same time, as part of an innate immune response, antimicrobial peptides, reactive oxygen species, and other epithelial secretions can enter the lumen and kill and digest the trapped bacteria [34]. Not all bacteria are killed, however, with species, such as *Pseudomonas aeruginosa* [28], *Salmonella enterica serovar* Typhimurium [29], and *Aeromonas caviae* [30,31], being able to be ingested and proliferate in the midgut before being shed in the feces in high numbers. The survival rate of ingested bacteria is dose-dependent [37] and also dependent on competition with the commensal microbiota [38]. Studies have shown that the numbers of pathogens in the digestive tract are three times higher than on the body surface, probably due to the multiplication of the pathogens in the digestive tract [39,40,41].

## 3. Transmission of Antimicrobial Resistance by Flies

### 3.1. Horizontal Transmission

It is now well known that bacterial pathogens with antimicrobial-resistant genes can be transmitted mechanically on the surfaces of flies, or in their feces, to new environments and animals/people [42]. Since 2009, however, there has been growing evidence that flies might be more than just mechanical vectors [43,44,45]. They might also act as vessels providing a suitable environment for ingested bacteria to transfer antimicrobial resistance genes to closely related bacterial species [43,44,45]. These can be shed in the feces and in this way spread antimicrobial resistance genes between different environments [46].

There is growing evidence the digestive tract of flies might serve as a suitable site for horizontal gene transfer [43,44,45]. Horizontal transfer of *tetM* on plasmid pCF10 among *Enterococcus faecalis* [44] and genes coding Shiga toxin and conferring antibiotic resistance in *E. coli* [45] have been demonstrated to occur in the gut of houseflies. In addition, an in vivo study showed that third-generation cephalosporin resistance genes, including *bla*CTX-M and *bla*CMY-2, were transferred successfully from *E. coli* to *Achromobacter* sp. and *Pseudomonas fluroresens* within the intestines of houseflies [43].

### 3.2. Vertical Transmission

There is also evidence that vertical transmission of antimicrobial resistance can occur in houseflies. When houseflies were fed with different concentrations of *Salmonella enterica*, *Cronobacter sakazakii*, *Escherichia coli* O157:H7, and *Listeria monocytogenes*, the organisms could be detected in the eggs of the next generation [47]. The same study demonstrated that *Salmonella enterica* and *Cronobacter sakazakii* fed to adult flies could be transmitted to the F1 [47]. A later study showed this can also occur with bacteria carrying antimicrobial resistance genes [48]. *E. coli*-containing plasmids with antimicrobial-resistant genes fed to houseflies were found in the subsequent immature and adult life stages [48]. In chickens, to which the immature stages were fed, the resistant *E. coli* persisted in the cecal contents for at least 16 days. There is thus growing evidence that houseflies are not only mechanical, but also biological vectors and, as such, they might facilitate the maintenance of antimicrobial-resistant genes [8,21,23,47,48]

## 4. Potential of Flies to Be Sentinels for Antimicrobial Resistance

Antimicrobial resistance remains one of the biggest threats to public health despite decades of efforts to lower the selection and transfer of resistance through more judicious use of antimicrobials [49]. There are more than two million illnesses and 23,000 deaths attributed to infections with antimicrobial-resistant bacteria every year in the USA [50]. In total, antibiotic resistance is estimated to add USD 20 billion annually to the direct healthcare costs in the USA, with additional costs to society resulting from lost productivity, which might be USD 35 billion a year [51].

Early detection of antimicrobial resistance in bacteria and ongoing surveillance are critical as they provide the information needed to monitor and develop therapy guidelines, infection control policies, and public health interventions [52]. Two surveillance systems have been commonly adopted: passive and active [53,54]. Although both can be used to monitor the prevalence of antimicrobial resistance in people and animals, they provide different ways to interpret surveillance and control strategies [54,55]. Passive surveillance involves the monitoring of resistance in routine samples collected from clinically ill patients [55,56]. It provides information on the current state of resistance in a naturally infected population [55,56]. In active surveillance, however, attempts are made to address a specific question by actively collecting data on resistance in defined infected and non-infected populations and locations [57]. Compared with passive surveillance, active surveillance is more labor intensive and costly and sample collection might be intrusive and involve ethical and personal issues [53,55]. Although the AMR surveillance approaches vary greatly in different countries, many unique phenotypes have been identified through the passive surveillance of pathogenic bacteria isolated from clinical specimens [53]. Furthermore, specific active surveillance programs for emerging resistant bacteria have been developed, for example for the ESBL-producing *Enterobacteriaceae*, methicillin-resistant *Staphylococcus aureus* (MRSA), and carbapenem-resistant Gram-negative bacteria [58].

Flies would appear to be useful surveillance vectors for tracking antimicrobial resistance. As mechanical vectors they can carry multiple bacterial pathogens and their resistance genes both externally and internally during all life stages [2]. Moreover, flies live in close association with people and their dwellings and can thus be easily and cheaply trapped for analysis. Methods for collecting flies include using aspirators [59], gauze traps over a bait [60,61], individual sweep/aerial nets [62,63], fly trap paper [64,65], jug traps [24,66], and the QuikStrike Abatement strip [67].

There are now many studies reporting on antimicrobial resistance genes in flies, reflecting the increasing awareness that they might play important roles in resistance transmission and maintenance [3,26,68,69]. Following an early study in 1983, in which nalidixic-resistant *Campylobacter* was reported in flies in Norway [70], there was little interest until 2005. Thereafter there have been numerous reports on a variety of bacteria and resistance genes in flies from Libya [19], North America [24,71,72,73,74], Morocco [75], Taiwan [26], Japan [76], the Netherlands [77], Spain [78], Zambia [79], Germany [80,81], Brazil [82,83], Bangladesh [84], Ethiopia [85], Thailand [68,86], Nigeria [69], and India [81] (Table 1). Flies were collected in multiple places, including hospitals, streets, abattoirs, poultry farms, cattle farms, pig farms, fish, and fast-food restaurants. Houseflies were the majority species during the collection and intestinal microbiota was frequently cultured in the collected flies against more than one antibiotic. Some bacteria harbor multiple antimicrobial-resistant genes, with the extended-spectrum β-lactamases and mobilized colistin resistance the most commonly observed.

## 5. Conclusions

Future studies, including epidemiological investigations and animal models, are warranted to explore the bacterial pathogens and resistance genes in the different fly species. Particularly, investigations need to be performed to accurately define if bacteria and resistance genes are effectively transmitted from flies to animals and people. The data from these studies are imperative to determine the definite roles flies might play in disseminating resistant bacteria and the threats flies pose to public health. As flies are ubiquitous, easy, and cheap to capture and process, the recent suggestion that they would seem to be ideal sentinels for antimicrobial resistance [74] warrants further investigation.

Over the past decades, there has been considerable research on the role flies might play in antimicrobial resistance. It is now known flies can carry a variety of bacteria and their antimicrobial resistance genes. Although commonly recognized as mechanical vectors, evidence is accumulating that flies are involved in the horizontal and vertical transmission of antimicrobial resistance genes. There is also some evidence that flies might be useful sentinels of antibiotic resistance in the environment and in animals and people. Further studies will more clearly define the roles played by flies in the transmission of antibiotic resistance between the environment, animals, and people and also on their usefulness as sentinels for resistance.

## Figures and Tables

**Figure 1 vetsci-09-00300-f001:**
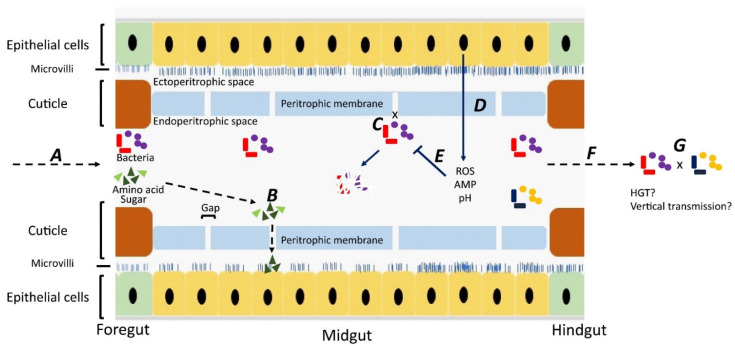
Fate of bacteria in the digestive tract (midgut) of flies. A: Ingested food with bacteria is predigested with saliva in the crop. The epithelial cells in the foregut are covered by a cuticle, which prevents bacterial invasion. B: In the midgut, digestion products can pass through gaps in the peritrophic membrane and enter the ectoperitrophic space to be absorbed by epithelial cells. C: Bacteria cannot pass through the gaps on the peritrophic membrane and remain in the endoperitrophic space. D: Bacteria are trapped in the endoperitrophic space, triggering an innate immune response in epithelial cells to produce reactive oxygen species (ROS) and antimicrobial peptides (AMP). E: Trapped bacteria are killed by ROS, AMP, pH changes, and digestive enzymes. F: Some bacteria survive in the hostile environment, pass through to the hindgut, and are shed. G: There might be horizontal gene transfer between bacteria surviving in the digestive tract and bacteria may be transmitted vertically to offspring.

**Table 1 vetsci-09-00300-t001:** Overview of multidrug antimicrobial resistance in flies.

Country and Sampling Period	Collection Sites	Flies	Bacteria	Resistant Gene	Resistant Antibiotics	Ref
Norway, 1983	Chicken and pig farms	Houseflies	*Campylobacter jejuni,* *Campylobacter coli*	N/A *	NAL	[70]
Libya, 2005	Hospital, streets, abattoir	Houseflies	*Pseudomonas,* *Staphylococcus aureus*	N/A	AMC, AN, CPH, D, K, NAL	[19]
USA, 2007	Chicken salad meal, chicken sandwich with French fries, carrot cake	Houseflies	*Enterococcus casseliflavus,* *Enterococcus faecium,* *Enterococcus faecalis,* *Enterococcus hirae*	N/A	TET, ERY, K	[71]
USA, 2008	Cattle feedlot, contaminated fast food	Houseflies	*Enterococcus faecalis*	N/A	TET, CIP, ERY	[72]
USA, 2009	Poultry farm	Houseflies, Blow flies, Bottle flies	*Enterococci.* *Staphylococcus aureus*	*erm(B)*, *erm(A)*, *msr(C)*, *msr(A*/*B)*	N/A	[24]
Morocco, 2010	Houses, garbage heaps, open defecating grounds	Houseflies	*Enterococcus*, *Escherichia coli,* *Klebsiella, Providencia,* *Staphylococcus*	N/A	AMG, Carbapenems	[75]
Taiwan, 2011	Pig farm	Houseflies	*Salmonella*	N/A	AMP, AMC, C, CIP, STREP, TET, NAL	[26]
Japan, 2013	Cattle barn	Houseflies, Stable flies	*Escherichia coli*	*blaCTX-M-15*		[76]
Netherlands, 2014	Poultry farm	Houseflies, Lesser houseflies, Stable flies, Blow flies	*Escherichia coli*	*bla*TEM-52	N/A	[77]
Spain, 2015	Poultry farm	Houseflies Lesser house fliesStable flies	*Escherichia coli*	blaCTX-M-1, blaCTX-M-14, blaCTX-M-9.	N/A	[78]
Zambia, 2016	Fish	Houseflies	*Escherichia coli* *Salmonella*	*blaTEM*, *blaSHV*	N/A	[79]
Germany, 2017	Pig farm	Stable flies	*Escherichia coli*		Colistin	[80]
Brazil, 2018	Cattle farm	Houseflies	*Escherichia coli*	*blaTEM*, *blaCTX-M*	N/A	[83]
USA, 2019	Dairy unit, dog kennel, poultry farm, beef cattle unit, urban trash facility and urban downtown	Houseflies Stable flies	*Escherichia coli, Staphylococcus* *Klebsiella pneumoniae,*	*bla*CMY-2, *bla*CTXM-1	N/A	[73]
Bangladesh, 2019	Hospital, livestock	Houseflies	*Escherichia coli,* *Salmonella*	N/A	AMG, ERY, OTC, TET	[84]
India, 2019	Livestock	Houseflies Blow flies	*Escherichia coli,* *Salmonella*	N/A	AMX, AMP, ATM, CTX, IPM	[81]
Germany, 2019	Hospital, household, zoo, streets	Houseflies Blow flies Flesh flies	*Acinetobacter, Citrobacter,* *Enterobacter, Escherichia coli,* *Klebsiella, Pseudomonas aeruginosa, Raoultella*	N/A	AMP, AMX, CEP, GEN, STREP, TET	[87,88]
Ethiopia, 2020	Hospital, market	Houseflies	*Klebsiella, Proteus*	N/A	C, GEN, SXT	[85]
Nigeria, 2020	Slaughterhouse trash, hospital	Houseflies	*Proteus, Salmonella* *Pseudomonas aeruginosa,*	N/A	AMX, AUG, GEN, STREP, SXT, TET	[89]
Thailand, 2021	Livestock	Blow flies	*Enterobacteriaceae Escherichia coli*	N/A	AMP, CEP, STREP	[68]
Brazil, 2021	Trash	Houseflies Blow flies	*Enterobacter, Escherichia coli,* *Klebsiella, Raoultella, Serratia,*	N/A	C, CTX, GEN, MEM, SXT, TET	[82]
USA, 2021	Livestock facilities	Houseflies Stable flies	*Bacillus, Commensalibacter,* *Enterococcus, Kytococcus,* *Oceanobacillus, Ochrobactrum*	N/A	CTX	[90]

AMC: amoxicillin/clavulanate (2:1); AMG: aminoglycoside; AMP: ampicillin; AMX: amoxicillin; AN Amikacin; ATM: aztreonam; AUG: augmentin; CEP: cephalosporin; CIP: ciprofloxacin, C: chloramphenicol; CPH: cephaloridine; CTX: cefotaxime; D: doxycycline; ERY: erythromycin; ESBL: extended-spectrum beta-lactamases; GEN: gentamicin; IPM: imipenem; K: kanamycin; MEM: meropenem; N: norfloxacin; NAL: nalidixic acid; OTC: oxytetracycline; STREP: streptomycin; SXT: trimethoprim-sulphamethoxazole; TEM: trimethoprim; TET: tetracycline. *: the information related to the resistant gene of antibiotics is not available.

## Data Availability

The data presented were obtained from all subjects involved in this study.

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
