# Peer review of "Flies as Vectors and Potential Sentinels for Bacterial Pathogens and Antimicrobial Resistance: A Review"

_vetsci, 2022, doi:10.3390/vetsci9060300_

Round 1

Reviewer 1 Report

Specific comments:

Abstract: Please review and rewrite the sentence on line 13-14.

Introduction section:

    1. It is necessary to include several references in this section that supports sentences (lines 21, 23, 27, 30, 32, 34 and 37).

Section 2: Flies as vectors of bacterial pathogens:

    1. It is necessary to include several references in this section that supports sentences (lines 48, 65, 70, 75 and 77)
    2. In lines 51-54 you talk about SARS-CoV-2 which is a virus and not a bacterial pathogen, so it is not the correct location for this information. You may include this information in the introduction section if you want.
    3. In line 78 please guarantee that all bacterial names are in italics.
    4. In line 81 change “microflora” to microbiota, that is the correct term.

Section 3 Transmission of antimicrobial resistance by flies:

    1. It is necessary to include several references in this section that supports sentences (lines 88, 91, 94, 96, 106, 107, 110 and 112)
    2. Please rewrite the sentence in lines 94-96. What do you mean by “genes related to Shiga Toxin and antibiotic resistance”?
    3. In line 97 you mention Third-generation cephalosporin resistance genes. What genes? Please include that information.

Section 4 – Potential of Flies to be sentinels for antimicrobial resistance:

    1. It is necessary to include several references in this section that supports sentences (lines 117, 124, 126, 127, 128 and 131);
    2. In lines 133 to 138 you mention the surveillance in human medicine, and refer that this surveillance is mostly passive, but this is not accurate, the surveillance is very diverse according to each country. Please improve this paragraph.

Section 5: Antimicrobial resistant bacteria and resistance genes reported in flies around the world

         1. Include reference line 150.

         2. Table 1 must be improved. A bacterium being resistant to an antibiotic is different from a bacterium where it was detected an antimicrobial resistant gene. So, in the Table 1 you have a column referred as “Resistance against” that must be changed. Be aware that antimicrobial resistance may be expressed genotypically and/or phenotypically. Also, phenotypically resistance to an antimicrobial may be caused by different genes. Please clarify this information in the table.

Conclusion section: Please rewrite sentence in line 166.

Figure 1: Please rewrite the legend of Figure 1, concerning Line 194-195. Also, the letter G described in the Figure does not appear in the image. Please verify that.

References section: Caution with the references!! All bacterial names must be in italics, as well as Musca domestica.

Author Response

Please find the PDF file as the response to reviewer-2's comments. Thank you!

Reviewer 2 Report

The review article "Flies as vectors and potential sentinels for bacterial pathogens 2 and antimicrobial resistance: a review" by Yin et al.,

The review is not organized well and does not provide any timeline and importance of the study. It should be considered as a mini-review as detailed descriptions are missing.

Section 2.1: External carriage of bacterial pathogens. This section should be elaborated on as currently, it cites only an example of E.coli infection.

Section 4: Potential of flies to be sentinels for antimicrobial resistance: This is a very general paragraph explaing in details about the antimicrobial resistance, whereas resistance development and spread were briefly presented in section 5.

Author Response

Please find the uploaded PDF file (responses to reviewer 2's comments). Thank you!

Reviewer 3 Report

The authors of the manuscript entitle “Flies as vectors and potential sentinels for bacterial pathogens and antimicrobial resistance: a review” aims to provide a review regarding the ability of flies to transmit bacteria and consequently antimicrobial resistance genes. Also, they describe the role of flies as sentinels for antimicrobial resistance.

 The manuscript addresses a scientifically significant topic and is well written. Table 1 should be divided in two tables: one with information regarding antimicrobials and the other regarding antimicrobial resistant genes.

Author Response

Dear Expert reviewer,

In following your advice, we have divided column-5 into two columns: resistant genes (column 5) and resistant antibiotics (column 6). The changes can be found in Table 1 on page -6 (with yellow highlight).

Thank you!

Reviewer 4 Report

Well made review useful for humans and animals in a real one health perpective. What  is surprising is the scarce news from Europe where there is a great hype and debate  about the effects of antibiotic resistance in farms and pets.

Author Response

Reviewer 3

Well made review useful for humans and animals in a real one health perspective. What is surprising is the scarce news from Europe where there is a great hype and debate about the effects of antibiotic resistance in farms and pets.

Response: Thank you for your kind comments. The first author of this review paper is in a PhD/Residency dual program, and she devotes herself to performing more research in the related areas. We thank you for your support and encouragement.